# Molecular Simulation to Explore the Dissolution Behavior of Sulfur in Carbon Disulfide

**DOI:** 10.3390/molecules27144402

**Published:** 2022-07-08

**Authors:** Xiangyu Cui, Wenbo Wang, Mengcheng Du, Delong Ma, Xiaolai Zhang

**Affiliations:** 1 School of Chemistry and Chemical Engineering, Shandong University, Jinan 250014, China; 202032358@mail.sdu.edu.cn; 2Shandong Yanggu Huatai Chemical Co., Ltd., Yanggu, Liaocheng 252300, China; wwb@yghuatai.com (W.W.); dumengcheng@126.com (M.D.); delong122@163.com (D.M.)

**Keywords:** molecular simulations, soluble sulfur, insoluble sulfur, solubility parameter theory

## Abstract

Soluble sulfur (S_8_) and insoluble sulfur (IS) have different application fields, and molecular dynamics simulation can reveal their differences in solubility in solvents. It is found that in the simulated carbon disulfide (CS_2_) solvent, soluble sulfur in the form of clusters mainly promotes the dissolution of clusters through van der Waals interaction between solvent molecules (CS_2_) and S_8_, and the solubility gradually increases with the increase in temperature. However, the strong interaction between polymer chains of insoluble sulfur in the form of polymer hinders the diffusion of IS into CS_2_ solvent, which is not conducive to high-temperature dissolution. The simulated solubility parameter shows that the solubility parameter of soluble sulfur is closer to that of the solvent, which is consistent with the above explanation that soluble sulfur is easy to dissolve.

## 1. Introduction

As the macromolecule of soluble sulfur (S_8_) polymerization [1,2], insoluble sulfur (IS) can replace soluble sulfur (S_8_) as a high-performance rubber additive. Insoluble sulfur has a slow migration speed in rubber, which can effectively prevent compounds from frosting in tire production and improve the adhesion between tire films. The corresponding tire products have excellent heat resistance and wear resistance. Therefore, insoluble sulfur (IS), as an indispensable raw material in the production of high-quality tires, especially radial tires, is a rubber vulcanizing agent and accelerator with broad development prospects [3,4].

CS_2_ is commonly used as the extractant of IS in industrial production, which can effectively separate common sulfur (soluble sulfur S_8_) and polymerized sulfur (insoluble sulfur IS) and is easy to recycle. Considering the volatility, flammability and explosion of CS_2_, the danger is high in the actual production process. Therefore, researchers at home and abroad are mostly looking for a new, effective extractant for IS. For example, Luo Hongyan [5] selected toluene as the extractant by comparing the extraction effects of trichloroethane, toluene and CS_2_ with their toxicity and operation requirements, and they extracted it at 80 °C for 45 min to obtain more than 90% IS products. Zhang Kejuan et al. [6] selected styrene with an aromatic ring and C=C bond to extract IS. The best operating conditions were a temperature of 77 °C, time of 10 min and liquid–solid ratio of 27, and the highest content was 94.2% and the extraction efficiency was 91.1%. The literature shows that no matter which extractant is selected, most of them are based on CS_2_ for comparison or as a part of the extractant for research.

As a useful supplement to experiments, molecular simulation can reveal the properties of soluble sulfur and insoluble sulfur at the molecular level. Jones et al. [7,8] studied the structure of S_n_ (*n* = 2–18) by density functional theory and the Monte Carlo method and considered that the number of sulfur atoms in the IS chain was around 100. Wang Rongjie et al. [9] put forward a ring-opening cracking mechanism for S_8_ and established a possible reaction path through the transition state theory. The product distribution obtained is similar to that obtained by the vapor density method. Ma Jian et al. [10] compared the thermal stability and thermodynamic properties of IS with different stabilizers on the micro level based on density functional theory, and the simulation results were consistent with the experimental results and DSC characterization results. These simulation calculations provide microscopic information for researchers to better understand IS and increase their cognitive understanding of IS.

In industrial production, in order to obtain a rubber accelerator with excellent performance, soluble sulfur S_8_ will be extracted at room temperature, and insoluble sulfur IS will be obtained, which will be used after being distinguished from each other [11]. Experimental workers must realize that the solubility of S_8_ in CS_2_ can be explained on a microscopic scale, and an understanding of the reasons for the solubility difference between S_8_ and IS is also helpful for industrial design in practical operation. In this paper, the molecular dynamics simulation method is used to study the dissolution behavior of S_8_ and IS in CS_2_ solvent at a molecular scale, in order to obtain the reason for the solubility difference between the two types of sulfur, explain the related dissolution phenomena and verify the feasibility of the IS model. As a toxic and harmful organic solvent, CS_2_ is not friendly to the environment. This work can also aid in the subsequent industrial production to find a new, effective and pollution-free extractant for IS.

## 2. Model Construction and Simulation Method

### 2.1. Model Construction 

Firstly, the monomolecular models of soluble sulfur (S_8_), insoluble sulfur (IS) and carbon disulfide (CS_2_) were constructed according to reference [12]. S_8_ is a planar regular octagonal structure, and IS is a polymer chain with 96 S atoms. It has been shown in the literature that ·S_8n_· is a metastable substance [13], and both ends of the polymerized sulfur molecular chain are free radicals. After polymerization, ·S_8n_· will break from both ends, so it is necessary to add a small amount of stabilizer to inhibit the breaking speed of the sulfur atom chain and improve the stability of IS. In this paper, an iodine atom is selected as the capping agent. After the structure is completed, the DMol3 module in Materials Studio (MS) software is selected, and the structure is optimized based on density functional theory. Geometry optimization is selected as the task; GGA and BLYP are set as the basis set and universal function, respectively, in the functional; charge is set to 0, and max. iterations is set to 200. The rest of the convergence tolerance can be selected by default, and the obtained optimized configuration of IS is shown in Figure 1. The S_8_ configuration is a three-dimensional crown shape [13], which conforms to its theoretical configuration. The optimized IS configuration is similar to the winding three-dimensional space configuration of “wool”.

### 2.2. Simulation Details

Based on the optimized structure, a CS_2_ solution model is constructed, as shown in Figure 2. We selected Gromacs 2019.4 software and the GROMOS54A7 [14] all-atom force field, and the force field parameters were derived from the Automated Topology Builder (ATB) tool [15]. For the soluble sulfur S_8_ system (model A), firstly, a certain number of S_8_ molecules (310) were placed in a box of 6 × 6 × 6 nm to minimize the energy, and then the isothermal and isobaric ensemble (NPT) simulation was carried out. The Berendsen hot bath method was adopted to control the temperature, the simulation step was 2 fs, and the total simulation time was 10 ns, so as to obtain the appropriate density. At the simulation endpoint, the system simulation density was approximately 2.015 g/cm^3^, and the relative error with the data from [16] was 2.18%. It could be proved that this method could be used to simulate the density of substances. The final box size was 4.32 × 4.32 × 4.32 nm. The optimized aggregation model was placed in the center of a new box with a size of 9 × 9 × 9 nm, filled with a certain number of CS_2_ molecules (6550) and then simulated by canonical ensemble (NVT) at 298 K, 318 K and 333 K, respectively, with a simulation step of 2 fs and a total simulation time of 30 ns. In the whole simulation process, the Velocity-Verlet algorithm was used to solve Newton’s motion equation for each particle. The LINCS algorithm was used to constrain the bond length, the Lennard–Jones potential function was used for Van der Waals interaction, and its truncation radius was 1.2 nm. The molecular dynamics trajectory was observed by the VMD program, and the simulation systems were called A-298, A-318 and A-333, respectively.

The insoluble sulfur (IS) system was simulated by a similar treatment method. Because the actual density of IS is unknown, the initial box of the NPT ensemble was built as 6 × 6 × 6 nm, and 10 IS molecular chains were placed in it. Finally, the simulated grid under the NPT ensemble was 4.005 × 4.005 × 4.005 nm. The IS system in CS_2_ solvent (the same the CS_2_ molecule added to the box to correspond to the above A model) had a box size of 9 × 9 × 9 nm and the simulation temperature was controlled at 333 K, which was called simulation system B-333.

## 3. Results and Discussion

### 3.1. Simulation Results

As can be seen in Figure 3a, at the simulation endpoint (30 ns) of system A-298, a small group of S8 molecules on the surfaces of S_8_ clusters will diffuse into the surrounding CS_2_ solvent, but the inner S_8_ molecules of the S_8_ clusters will also move from the inside to the spherical shell of the S_8_ clusters, and the diameter of the S_8_ clusters is around 4 nm at this time. It can also be seen from the figure that the S_8_ cluster density decreases slightly, and its centroid also moves in the CS_2_ solvent. According to Figure 3a (A-298), b (A-318) and c (A-333), when the endpoint equilibrium is simulated, with the increase in temperature, more S_8_ molecules from the surfaces of the S_8_ cluster will diffuse into the surrounding CS_2_ solvent, more S_8_ molecules from the interior of the S_8_ cluster will move into the spherical shell of the S_8_ cluster, and the diameter of the S_8_ cluster will be altered. Comparing the conformation of the last frame of the model in Figure 3d (B-333) with its initial structure at the simulation endpoint, it is found that the centroid of the IS cluster will shift in the box space, there is no dissolution trend, and its volume in the space has no obvious change.

### 3.2. Solubility Analysis of Soluble Sulfur

#### 3.2.1. Properties of Dissolved Sulfur

In order to better study the relationship between S_8_ molecules and S_8_ clusters in the simulation process, the number density change of S_8_ in solution was studied in this paper. As shown in Figure 4b, taking the center of mass of the S_8_ cluster as the center, the number of S_8_ molecules contained from radius R to r + dr was counted as N_S(r)_ [17], and the number density of S_8_ molecules in the “spherical shell” was counted by formula (1). When the radius of the S_8_ cluster tends to be stable, the distance between each S_8_ molecule and the centroid of S_8_ cluster was calculated.
(1)$$\rho_{S}\left(r\right)=\frac{N_{S\left(r\right)}}{\frac{4}{3}\pi \left[{\left(r+dr\right)}^{3}-r^{3}\right]}$$

As shown in Figure 4b, the whole space can be divided into three areas with increasing distance from the center of mass: the S_8_ cluster area, transition area and CS_2_ area. Near the centroid of the S_8_ cluster, *ρ*_S(r)_ tends to be stable, which is the S_8_ cluster area. In the transition region, the density distribution *ρ*_S(r)_ decreases rapidly with the increase in the distance from the centroid, and this region is occupied by different numbers of CS_2_ and S_8_ molecules. When *ρ*_S(r)_ decreases to a stable value, the S_8_ molecule at the stable value is considered to have entered the CS_2_ region. Moreover, in the transition region, with the increase in the distance from the centroid, the number density of S_8_ molecules decreases by an order of magnitude compared with that of the S_8_ cluster region.

In molecular dynamics simulation, the “10-90” method [18] is often used to determine the boundary between two phases. In order to better determine the boundary between the two phases, the trajectories of models A-298, A-318 and A-333 were analyzed. As shown in Figure 4a, ρ_S(r)_ was counted and the position of the junction was observed. It is found that ρ_S(r)_ in the figure has an obvious downward trend at a certain position; that is, the dotted lines in the figure are, respectively, defined as the junction of the S8 cluster area and transition area. According to this characteristic, the x coordinate value of each line segment is defined as m. For the three systems, A-298, A-318 and A-333, when the m value of the center of mass of the S_8_ molecule in the system is greater than 10, 15 and 18, respectively, it is considered that the S_8_ molecule is dissolved in CS_2_, and S_8_ is considered to be dissolved in CS_2_ at this time.

According to the definition in this paper, the dissolved sulfur is expressed as a molar fragment (M.F.). It can be seen from Figure 5 that the M.F. changes with the simulation time, but gradually approaches a stable value, which is close to the experimental value (the solid line parallel to the X axis in the figure). Figure 5a–c are schematic diagrams of the M.F. of A-298, A-318 and A-333 with simulation time, respectively. It can be clearly seen that with the increase in temperature, the molar ratio of S_8_ to CS_2_ gradually increases, and the number of dissolved S_8_ molecules also gradually increases, until the system reaches equilibrium; it will tend to a stable value, and this value is almost consistent with the data in [19]. As shown in Figure 5a,c, it takes around 22 ns for M.F. to converge at 298 K, but only around 15 ns at 333 K. This shows that only from these three temperatures, with the increase in simulated temperature, M.F. can converge faster. From the production perspective, a higher temperature can indeed improve the extraction efficiency, but it will also cause more energy consumption. According to this simulation, 318 K (45 °C) may be a good choice for production.

#### 3.2.2. Dissolved Sulfur Interaction

Noncovalent interactions (NCI) analysis [14] is a tool that is generally applicable to graphically display the properties and occurrence areas of intermolecular noncovalent interactions. In order to better show the weak interaction between S8 and CS_2_, the independent gradient model (IGM) analysis [20] of these compounds was carried out via Multiwfn software [15]. The aRDG method is used to study the type and intensity of a weak interaction between a single S_8_ molecule and CS_2_, and the weak interaction intensity is generally measured by interaction energy. In AIM theory, *ρ*_(r)_, the critical point of weak interaction, is one of the most important indexes to measure the strength of interaction. Its value has a positive correlation with the strength of the bond, so it is also used to define the bond level. By mapping the value of *ρ*_(r)_ to the RDG isosurface in different colors, the aRDG diagram can be obtained, and the intensity of its interaction is clear at a glance. As shown in Figure 6, the bluer the color is, the stronger the electrostatic and hydrogen bond effects are; the redder the color is, the more obvious the steric hindrance effect is; meanwhile, the green color indicates that the average density value of the corresponding position is lower, which corresponds to a weak effect, namely the dispersion effect. The whole S_8_ forms a green isosurface, which clearly shows that S_8_ molecules tend to form van der Waals interactions with other CS_2_ molecules in these directions, so van der Waals interaction is the dominant factor affecting the weak interaction between S_8_ and CS_2_ molecules [21].

It is found that S_8_ molecules located at the edges of clusters can be dissolved in CS_2_ solvent under the interaction of CS_2_ molecules. We selected an S_8_ molecule in the subsurface of the S_8_ cluster and analyzed its trajectory. From Figure 7, it can be seen that in the first ns, a single S_8_ molecule will interact with its surrounding S_8_ molecules, continuously rotate and adjust its position in space in the sub-surface area of the S_8_ cluster, and it will then gradually move to the surface area of the S_8_ cluster. At around the ninth ns, the molecule will gradually escape the control of the S_8_ cluster and come into contact with more CS_2_ molecules. Then, under the action of van der Waals force, it will rotate and adjust its position in space until it completely enters the CS_2_ phase.

### 3.3. Solubility Analysis of Insoluble Sulfur

#### 3.3.1. Dissolution Behavior of Insoluble Sulfur

It can be seen from the above that S_8_ can be dissolved in CS_2_, and the amount of dissolved sulfur increases with the increase in temperature in a certain temperature range, but it is not soluble in CS_2_ for polymeric sulfur IS. First of all, the simulation results of IS in the CS_2_ system (system B-333) can be seen from the last image in Figure 3. In the last frame of the simulation, the IS cluster is not dispersed, and it can be seen from its mass density distribution (as shown in Figure 8b) that there is only one peak in the whole simulation process from the beginning of the simulation to the end of the simulation. Combined with the IS cluster in Figure 8a, its volume expands slightly in space with the increase in simulation time. Furthermore, it can be seen from Figure 8b that the mass density curve of the 30 ns represented by the green curve is slightly lower and larger than the peak of the mass density curve of the 0th ns represented by the black line, which is in good agreement with the experimental results. It can be seen that IS clusters do not dissolve in the CS_2_ system, but their clusters will change in different positions in the CS_2_ system space.

#### 3.3.2. Solute Interaction with Solvent

In order to determine why S_8_ can be dissolved in CS_2_ and IS cannot be dissolved in CS_2_ at the microscopic level, firstly, the interaction energy images of solute molecules S_8_ and IS and solvent molecules CS_2_ in systems A-333 and B-333 with simulation time were generated. From Figure 9, it can be seen that the interaction energy between IS and CS_2_ represented by the red curve at the top fluctuates in the equilibrium range of −5000 KJ/mol within the simulation time of 30 ns, which indicates that IS clusters are only in the CS_2_ solvent. The interaction energy curve of S_8_ and CS_2_ represented by the black curve at the bottom of the whole shows that the interaction energy decreases from around −12,000 kJ/mol to around −27,500 kJ/mol at the beginning of the simulation, until it fluctuates around −27,500 kJ/mol, indicating that S_8_ quickly dissolves in CS_2_ in the first time period, and then from approximately 15,000 ps; gradually, the system reaches equilibrium, and some undissolved S_8_ still exists in the system, which is not completely dissolved, in line with the above conclusion.

### 3.4. Solubility Parameter Theory

Through the molecular dynamics simulation, it can be concluded that S_8_ molecule IS dissolved in CS_2_ through van der Waals interaction, but IS was insoluble. We can also explain this from the perspective of solubility parameter theory [22,23,24]. The concept of the solubility parameter was first put forward by Hildebrand [25], and it is an important parameter to characterize the interaction strength of simple liquid molecules. At present, solubility parameters are widely used in the calculation of the phase equilibrium of multi-component systems, solvent extraction, selection of coating solvents and other fields. In the research of solubility, both low-molecular substances and high-molecular substances are applied in their condensed state. This aggregation state is determined by the geometric arrangement of molecules, which determines the physical properties of substances. The force between molecules in the aggregated state is not the chemical bond force, but the force between unbonded atoms, which can be divided into van der Waals force dispersion force, induction force, couple force and hydrogen bond. Usually, cohesive energy, cohesive energy density or solubility parameters are used to express intermolecular forces. The solubility parameter is defined as the square root of cohesive energy density.
(2)$$\delta ={\left(\frac{E}{V}\right)}^{0.5}={\left(\frac{U_{m}}{V_{m}}\right)}^{0.5}$$

In Equation (2), E is cohesive energy (KJ/mol), V is volume (ml/mol), and E and V are the cumulative values of the energy and volume of each group constituting the molecule, respectively; U_m_ is the molar evaporation energy of the polymer, and V_m_ is the molar volume of one repeating unit of the polymer [26]. It is suggested that a carrier with a smaller difference (ΔSP) from the solvent solubility parameter should be selected as far as possible when screening carriers. If the solubility parameters of the solute and solvent are similar, they are easily mutually soluble [27]. Before studying the solubility parameters of IS, it is necessary to verify the feasibility of the simulation of the Focite module in MS software. At this time, task is selected as Cohesive Energy Density, the forcefield is set to Compass, and charges is set to forcefield assigned. Electrostatic and van der Waals are controlled by the atom-based method. The known solubility parameters of benzene (C_6_H_6_), ethyl acetate (CH_3_COOC_2_H_5_) and carbon disulfide (CS_2_) are calculated first.

It can be seen from Table 1 that the relative errors between the simulated values of the solubility parameters of C_6_H_6_, CH_3_COOC_2_H_5_ and CS_2_ and the reference values in the literature are all below 10%. Therefore, it can be considered that the above-mentioned methods for simulating the solubility parameters can be applied to predict the solubility parameters of a substance. Then, the solubility parameters of S_8_ and IS are calculated by using the above method of calculating solubility parameters and the aforementioned molecular models of S_8_ and IS, and the simulated values of solubility parameters of S_8_ and IS are shown in Table 2.

The literature shows that the ΔSP of solute and solvent in solid dispersion is 1.6−7.5. When ΔSP is melted at 1.6–7.5, solute and solvent are completely miscible; when ΔSP is in liquid state at 7.4−15.0, they are completely immiscible when ΔSP > 15.9 [29]. According to the above theoretical calculation, the difference ΔSP in solubility parameters between S_8_ and CS_2_ at 333 K is 8.256, ranging from 7.4 to 15.0, and they are partially soluble, while the difference ΔSP in solubility parameters between IS and CS_2_ is 16.953, which is greater than 15.9, and they are completely immiscible, which accords with this theory. It can be concluded that the IS model constructed above is reasonable and can play a role in the future study of IS. However, solubility parameters cannot fully reflect the influence of the molecular structure and spatial structure of the solvent on sulfur solubility, so it is not comprehensive to judge whether a solvent is suitable or unsuitable for sulfur solubility simply by the size of solubility parameters.

## 4. Conclusions

In this paper, the MD simulation method was used to study the dissolution behavior of S_8_ and IS in the CS_2_ system. From the whole simulation process, it is in line with the experimental expectation, so the model can be applied to explore the sulfur and insoluble sulfur problems in a microscopic way. From the simulation process, it can be seen that some S_8_ molecules around S_8_ clusters will diffuse into the CS_2_ system until the system reaches equilibrium, and the radius of S_8_ clusters will gradually decrease and become stable. As for the IS cluster, its position fluctuates in space, but there is no sign of scattering. We also introduced the definition of dissolved sulfur into the analysis of S_8_ in dissolution in the CS_2_ system. Through the simulation at 30 ns, we can see that S_8_ can dissolve in CS_2_, and its dissolution efficiency is different at 298 K, 318 K and 333 K. The higher the temperature, the better the dissolution effect. Moreover, from the industrial production point of view, the higher the temperature, the higher the dissolving efficiency of S_8_ in CS_2_, but at the same time, more energy and resources will be wasted. From the microscopic level, judging the weak interaction between solute S_8_ and solvent CS_2_ by the aRDG method, it can be found that van der Waals interaction is the dominant factor affecting the weak interaction between S_8_ and CS_2_. Furthermore, the solubility behavior of S_8_ and IS in the CS_2_ system was explained from the point of view of their interaction energy. In addition, the solubility parameter theory was used to verify that the IS model is reasonable, and the solubility parameter difference ΔSP between IS and carbon disulfide at 333 K is 16.953, which is greater than 15.9, and they are completely immiscible, which accords with this theory, and thus our findings can provide a reference for future research on IS problems, especially for finding a new, effective and pollution-free extractant as an alternative to CS_2_. However, the solubility parameter cannot fully reflect the influence of the molecular structure and spatial structure of the solvent on the solubility of sulfur, so it is not comprehensive to judge the solubility of a solvent for sulfur simply from the solubility parameter; however, the solubility parameter method can be applied to predict whether a substance is soluble in another substance.

## Figures and Tables

**Figure 1 molecules-27-04402-f001:**
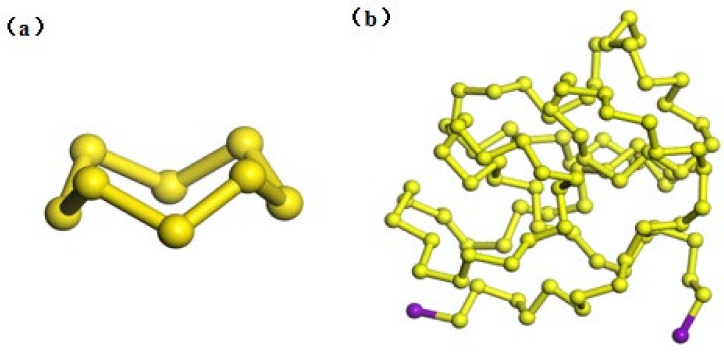
Chemical structure of two single molecules: (**a**) S_8_, (**b**) IS.

**Figure 2 molecules-27-04402-f002:**
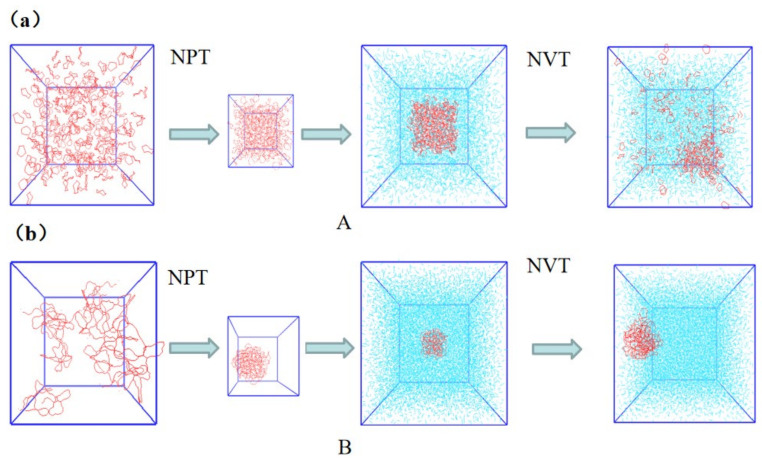
Model building process for (**a**) S_8_ and CS_2_ system, (**b**) IS and CS_2_ system.

**Figure 3 molecules-27-04402-f003:**
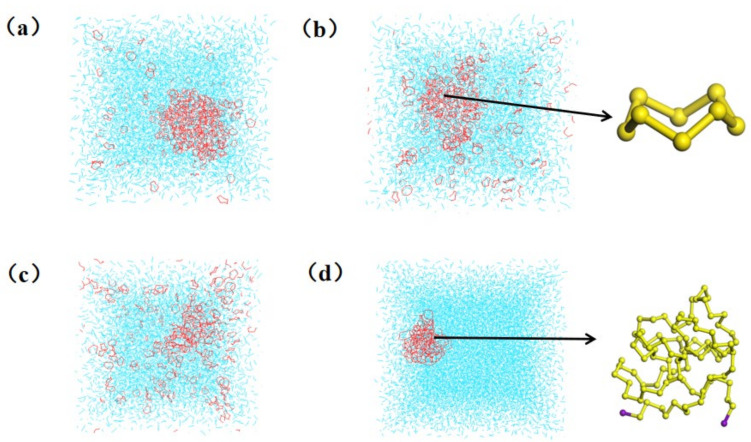
Each system of model A and B simulates the last frame conformation for (**a**) A-298 model, (**b**) A-318 model, (**c**) A-333 model, (**d**) B-333 model.

**Figure 4 molecules-27-04402-f004:**
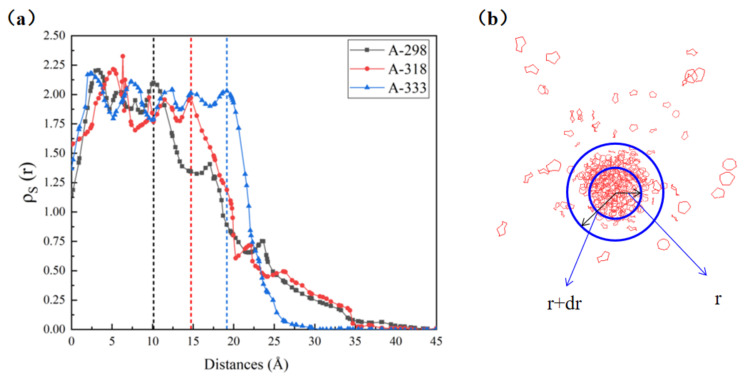
(**a**) The variation trend of *ρ*_S_(r) of models A-298, A-318 and A-333 with radius r; (**b**) position projection of S_8_ cluster on X-Y plane during simulation (CS_2_ molecule is not shown).

**Figure 5 molecules-27-04402-f005:**
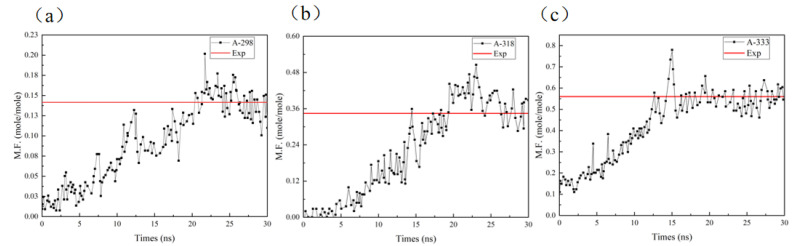
Schematic diagram of M.F. change with time for (**a**) A-298 model, (**b**) A-318 model, (**c**) A-333 model.

**Figure 6 molecules-27-04402-f006:**
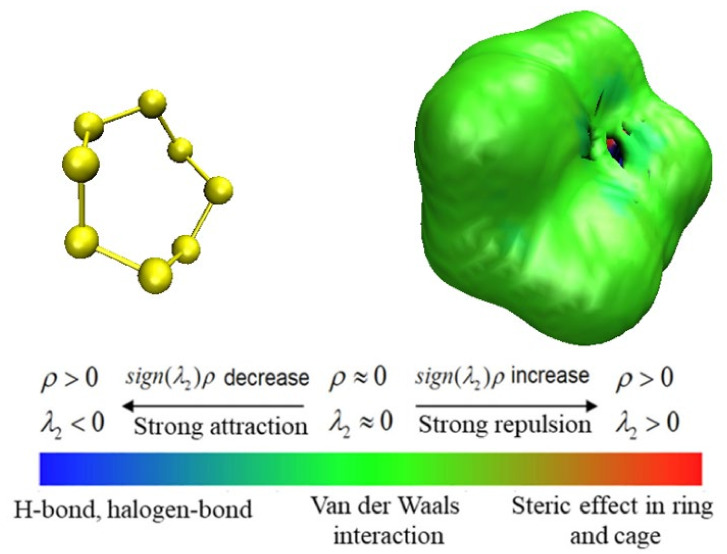
Schematic diagram of average reduced density gradient (aRDG) of single S_8_ molecule in CS_2_ system.

**Figure 7 molecules-27-04402-f007:**
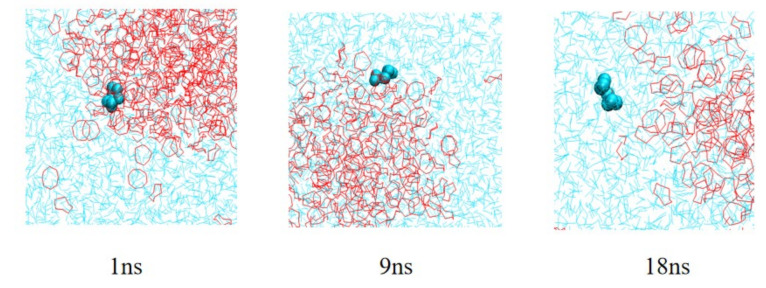
Partial enlargement of single S_8_ molecule dissolved in CS_2_ system (cyan is single S_8_ molecule, light blue is CS_2_ molecule).

**Figure 8 molecules-27-04402-f008:**
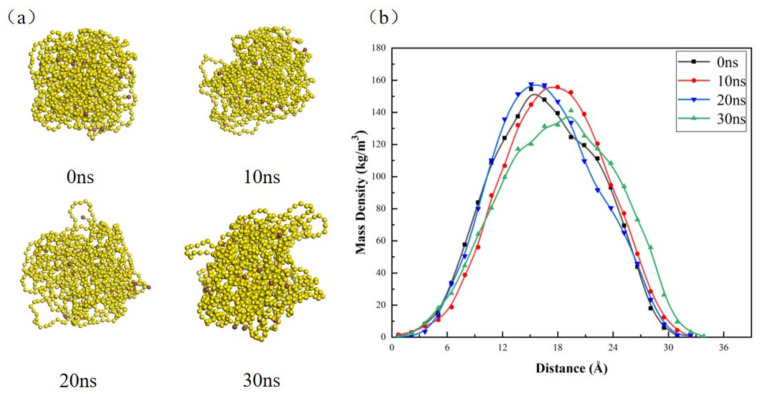
(**a**) The change diagram of IS cluster with simulation time in B-333 model (CS_2_ molecule is omitted); (**b**) the graph of mass density of IS clusters with simulation time.

**Figure 9 molecules-27-04402-f009:**
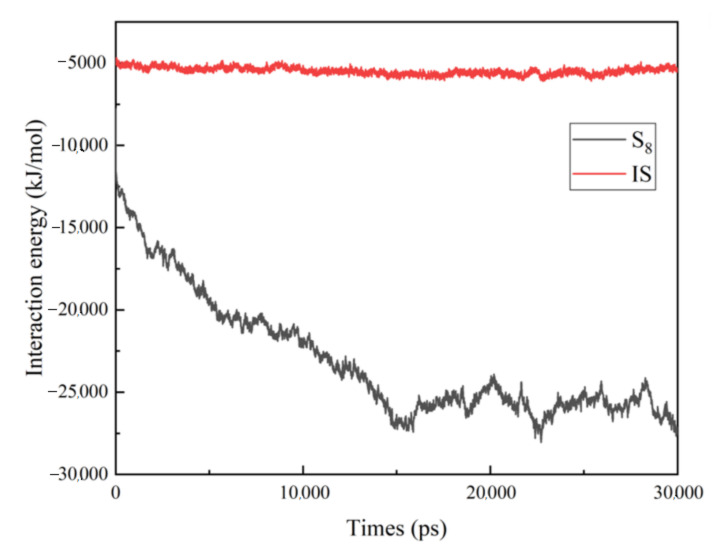
Interaction energies of solute molecules S8 and IS with solvent molecules CS_2_ in model A-333 and model B-333 with simulated time.

**Table 1 molecules-27-04402-t001:** Simulation value of solubility parameters of each substance and literature value [28] (P_0_ = l atm, 296.15 K).

Substance Name	Simulation Value/(MPa1/2)	Reference Value/(MPa1/2)	Relative Error/%
C_6_H_6_	17.6	19.5	9.7
CH_3_COOC_2_H_5_	17.4	18.8	7.4
CS_2_	20.6	19.4	5.8

**Table 2 molecules-27-04402-t002:** Solubility parameter simulation value of S_8_ and IS (P_0_ = l atm, 333 K).

Substance Name	Simulation Value/(MPa^1/2^)
S_8_	29.943
IS	38.631
CS_2_	21.678

## Data Availability

Data presented in this study is enclosed in the manuscript.

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
