# Peer review of "Molecular Simulation to Explore the Dissolution Behavior of Sulfur in Carbon Disulfide"

_molecules, 2022, doi:10.3390/molecules27144402_

Round 1
Reviewer 1 Report
The authors studied solubility of two sulfur-based materials in CS2 using molecular dynamics. Practically, this research is meaningful and interesting. However, the overall quality of this manuscript is not satisfying. The authors are urged to modify the manuscript carefully. Detailed comments/questions are listed below.
What does it mean by “Max. interaction is set to 200” on page 3?
why to choose 96-S as a representative of an insoluble sulfur material? Does the chain length have effects on the solubility?
Why to use NVT instead of NPT for the mixtures? Why the authors expected the volume of the system does not change?
To convince the validity of the simulation setups, the authors should at least report the density of each individual substance and compare with the available experimental data available in the literature.
Is a total duration of ’30 ns’ long enough for the systems to reach equilibrium?
In the calculation of RDF (radial distribution function), I don’t see the point to report and discuss about the three critical values of r (i.e. 10, 15, and 18 A, respectively), which merely are transient parameters varying with many effects such as time, concentration, temperature, and so on. Based on those numbers, I cannot get any useful information.
Please clarify the definition of ‘molar ratio’, and why the abbreviation is ‘MF’? Also, the authors should specify the references from which the experimental data was extracted.
Aside from reporting RDF and MF, the authors should also report molecular diffusion (rate) to clarify the effect of temperature.
Regarding the last sentence on page 7, if the charges on S8 are set to zero, there will be no other type of interactions (Coulomb) but pairwise vdW.
What exactly does ‘interaction energy’ refer to?
Please specify the definition of ‘solubility parameter’.
References should be given for the experimental values reported in Table 1.
Why the simulated solubility parameters of CS2 in Table 1 and 2 are different?
To me, S8 is fully miscible in CS2, why the solubility parameter falls in the range of partially soluble? What happens if you run the S96 case sufficiently long? Again, ’30 ns’ is too short to dissolve a long-chain molecule.
After reading through the manuscript, I still don’t have a clear mind about the mechanism. Given the same non-bonded interaction between sulfur atom in S8 and S96, it is the aromatic molecular structure prohibit the accumulation of S8 molecules and dissolve easily in CS2, whereas S96 tends to coil up and does not allow CS2 molecules to sneak in?
Reviewer 2 Report
This manuscript reports a study about a molecular simulation to explore the dissolution behavior of sulfur in carbon disulfide. Specifically, molecular dynamics simulation was applied to discuss soluble sulfur (S8) versus insoluble sulfur (IS). Detailed solubility differences and solubility parameters were evaluated based on the theoretical model. Overall, this report presents a comprehensive work though some details might be questionable. This has somewhat deteriorated the technical merit of this work. Some specific comments are provided below for the authors’ consideration to improve this manuscript before its publication.
1. The author gives a brief introduction of this work in abstract, but it would be better to present in a more organized way and further highlight the merits. For example, the author may want to briefly address the following points: 1. The importance of the problem; 2. The novelty of this study and the method applied in this work; 3. The major discoveries/conclusions.
2. The author mentioned the importance of simulation to experiment and industry; however, the simulation results were barely related to other’s experimental observations.
3. The author compared the simulation results of H2O, C2H5OH, and CS2, versus ref results, to support the reliability of simulated results. However, the simulation of soluble sulfur or polymerized sulfur has much more complicated conditions than the aforementioned molecules with simple structures, thus it is more convincing to provide the ref of S8 or IS from experimental results or previous publications.
4. The author highlighted the temperature influence on dissolution efficiency etc., which was barely discussed in the results discussion. Also, it is better to organize the conclusion and highlight the following points: 1. The problem addressed in this paper; 2. The developed method and the advancement; 3. The major improvement/observations/conclusions; 4. The importance of this work/future applications.
5. The author shall pay more attention to font and details, such as the first paragraph, equations, ref 6, etc.
Round 2
Reviewer 1 Report
The authors addressed most of my comments/questions well. Following list a few further comments:
- The authors should specify the reference(s) from where the experimental results were extracted in the caption of Fig.5
- Regarding Fig.R3, there are details of how the results were obtained? Are they fitting curves? I don't get how the "temperature effect" was addressed from this figure.
- If there was no Coulomb interaction, the authors should remove the corresponding words from the text.
- If the experimental references cannot be found, either the authors do not report experimental results, or the authors keep finding. It is dishonest to report values without giving correct references.
Reviewer 2 Report
The author has revised the manuscript accordingly
Author Response
Thank you for your comment and your recognition of our work.